# FEW-SHOT HYBRID DOMAIN ADAPTATION OF IMAGE GENERATOR

**Hengjia Li**[*1,4]    **Yang Liu**[*2]    **Linxuan Xia**[1]    **Yuqi Lin**[1]    **Wenxiao Wang**[†3]

**Tu Zheng**[†4]    **Zheng Yang**[4]    **Xiaohui Zhong**[5]    **Xiaobo Ren**[6]    **Xiaofei He**[1,4]

[1] State Key Lab of CAD&CG, Zhejiang University [2] Alibaba Cloud  [3] Zhejiang University
[4] Fabu Inc.  [5] Ningbo Beilun Third Container Terminal Co., Ltd  [6] Ningbo Zhoushan Port Co., Ltd

## ABSTRACT

Can a pre-trained generator be adapted to the hybrid of multiple target domains and generate images with integrated attributes of them? In this work, we introduce a new task – Few-shot *Hybrid Domain Adaptation* (HDA). Given a source generator and several target domains, HDA aims to acquire an adapted generator that preserves the integrated attributes of all target domains, without overriding the source domain's characteristics. Compared with *Domain Adaptation* (DA), HDA offers greater flexibility and versatility to adapt generators to more composite and expansive domains. Simultaneously, HDA also presents more challenges than DA as we have access only to images from individual target domains and lack authentic images from the hybrid domain. To address this issue, we introduce a discriminator-free framework that directly encodes different domains' images into well-separable subspaces. To achieve HDA, we propose a novel directional subspace loss comprised of a distance loss and a direction loss. Concretely, the distance loss blends the attributes of all target domains by reducing the distances from generated images to all target subspaces. The direction loss preserves the characteristics from the source domain by guiding the adaptation along the perpendicular to subspaces. Experiments show that our method can obtain numerous domain-specific attributes in a single adapted generator, which surpasses the baseline methods in semantic similarity, image fidelity, and cross-domain consistency. Project page is at `https://echopluto.github.io/FHDA-project/`.

## 1 INTRODUCTION

Few-shot generative *domain adaptation* (DA) aims to adapt a pre-trained image generator (Karras et al., 2019; Brock et al., 2018; Vahdat et al., 2021; Rombach et al., 2022) from the source domain to a new target domain using only few reference images. Existing methods (Mo et al., 2020; Li et al., 2020; Xiao et al., 2022; Mondal et al.) generally seek to achieve realistic and diverse generation which acquires salient characteristics of the target domain and preserves the variations learned from the source domain. Building upon previous promising progress toward DA, it is straightforward to derive the adapted model given images from *sketch* domain as shown in Fig. 1 (Left). However, a challenge remains when we require to generate images with integrated attributes of *sketch*, *smile*, and *baby*, given real images from individual domains. Besides, in real-world scenarios, images from the hybrid domain (*e.g.*, a *smiling baby* with the style of *sketch*) tend to be more difficult to collect compared with single domain. Under such circumstances, conventional DA becomes less feasible.

Instead of DA, we investigate this novel task – Few-shot generative *Hybrid Domain Adaptation* (HDA). Given a source generator and few-shot images of several target domains, HDA aims to acquire an adapted image generator that preserves the integrated attributes of all target domains, without overriding the original characteristics. Compared with DA, HDA offers greater flexibility and versatility to adapt generators to more composite and expansive domains. For example, given the

---

* Equal contribution; † Corresponding authors.

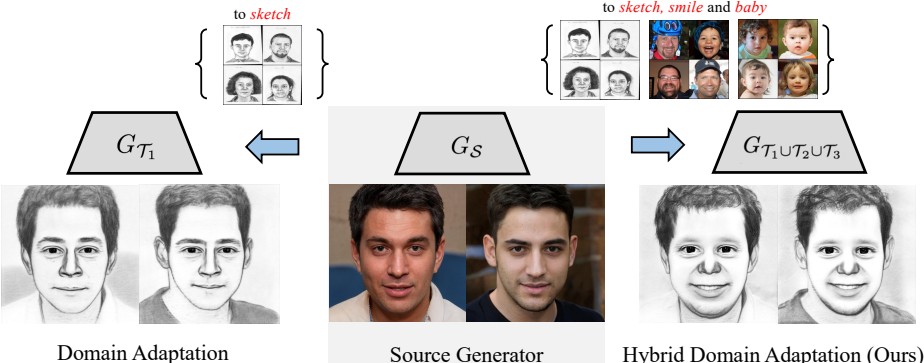

Figure 1: **(Middle)** Given a source generator ($G_{\mathcal{S}}$) pre-trained on a large-scale dataset, we propose to adapt it to the new target domain. **(Left)** *Domain Adaptation* adapts the generator from the source domain (*human(sketch)*). The adapted model $G_{\mathcal{T}_1}$ captures the target distribution using extremely few-shot references. **(Right)** Our newly introduced *Hybrid Domain Adaptation* aims to adapt to $G_{\mathcal{T}_1 \cup \mathcal{T}_2 \cup \mathcal{T}_3}$ that generate images with integrated attributes from $\mathcal{T}_1$ (*sketch*), $\mathcal{T}_2$ (*smile*), and $\mathcal{T}_3$ (*baby*), given few-shot references from multiple individual domains.

few-shot references from the individual *sketch*, *baby* and *smile* domains as shown in Fig. 1 (Right), HDA aims to adapt the generator to *sketch-smile-baby* domain. Concurrently, these images retain the primary characteristics of the source domain, which upholds cross-domain consistency.

Compared with conventional DA, HDA is more challenging in two aspects: (1) Existing DA approaches typically employ the discriminator to discern whether generated images belong to the target domain. However, we only have images sampled from individual domains and lack real images of the hybrid domain, which presents challenges for designing the discriminator-based adaptation framework. (2) Since there are extremely few reference images, the discriminator could easily overfit the easy-to-learn characteristics of certain target domains (Ojha et al., 2021), leading to missing the characteristics from other target domains in the generator.

To solve these issues stemming from the reliance on discriminator, we propose a discriminator-free framework for HDA. Instead of directly distinguishing whether generated images are real or fake, we endeavor to project distinct domains into separate embedding subspaces. Inspired by image classification task (Dosovitskiy et al., 2020; Oquab et al., 2023; Liu et al., 2021) that encodes images into high-dimensional embedding space and extract well-separable features, we posit that the pre-trained image encoder intrinsically functions as a hybrid domain classifier. Leveraging this property, we utilize these well-known pre-trained encoders to embed different domains' images into several distinct embedding subspaces.

Employing these well-separable subspaces, we introduce the directional subspace loss comprised of a distance loss and a direction loss to achieve HDA. Specifically, our objective is for generated images to progressively approach all the target subspaces. To this end, we propose the distance loss to minimize the distances from the generated images to all target domains' subspace. Solely employing the distance loss can result in model collapse, since distinct generated images may project onto the same point on the subspace and compromise cross-domain consistency. To preserve more characteristics from the source domain, we further propose the direction loss to guide the adaptation along the perpendicular to subspaces. Additionally, we can easily use our directional subspace loss for single domain adaptation since DA is a special form of HDA.

We validate the effectiveness of our method across multiple target domains. Compared with other potential approaches for achieving hybrid domain (Wu et al., 2023; Gal et al., 2021), our method accomplishes the fastest adaptation without training models on multiple individual domains. For both DA and HDA, our method achieves superior semantic similarity with the target domain, better image fidelity, and more characteristics preservation of the source domain compared with existing approaches. Overall, our contributions are summarized as follows:

- We introduce a novel task – Hybrid Domain Adaptation (HDA). Compared with DA, HDA offers greater flexibility and versatility to adapt generators to more composite and expansive domains.

- We propose a novel discriminator-free framework with directional subspace loss for HDA. Compared with other potential approaches, our methods accomplish fast and versatile adaptation without training models on multiple individual domains.
- For both DA and HDA, our method surpasses existing methods on the semantic similarity with the target domain, image fidelity, and cross-domain consistency. Qualitative and quantitative results demonstrate the effectiveness of our method.

## 2 RELATED WORKS

**Few-shot Generative Domain Adaptation.** Few-shot generative domain adaptation aims to adapt a pre-trained image generator to a new target domain with a limited number of reference images. Due to the scarcity of training images, existing works (Mo et al., 2020; Li et al., 2020; Ojha et al., 2021; Zhao et al., 2022b; Xiao et al., 2022; Mondal et al.) often adopt extra regularization terms to avoid overfitting. For example, CDC (Ojha et al., 2021) proposes the instance distance consistency loss to preserve the distance between different instances in the source domain. RSSA (Xiao et al., 2022) proposes a relaxed spatial structural alignment method to preserve the spatial structural information of the source domain. AdAM (Zhao et al., 2022a) proposes Adaptation-Aware kernel Modulation to address general FSIG of different source-target domain proximity. While previous works achieve promising progress toward generative domain adaptation, they rely fundamentally on the discriminator which makes it difficult to handle hybrid domain adaptation.

**Inference Time Interpolation for Hybrid Domain Adaptation.** Taking advantage of the disentanglement in the latent space of StyleGAN (Härkönen et al., 2020; Shen & Zhou, 2021; Xu et al., 2022; Shen et al., 2020; Wu et al., 2020), some works (Wu et al., 2023; Nitzan et al., 2023; Gal et al., 2021) are capable of producing images from the hybrid domain through the interpolation technique. For example, DoRM (Wu et al., 2023) adapts the source generator to each domain individually and then interpolate multiple domain's latent codes at inference time to generate images from the hybrid domain. However, the approach of inference time interpolation needs to train models on multiple individual domains, which necessitates multiple times the model size and training time. In contrast, our method can acquire one model for the hybrid domain in just few minutes.

**Pre-trained Image Encoder as Implicit Domain Classifier.** Image encoder has been extensively explored to extract salient and representative features, which typically projects the images into a well-separable embedding space for classification. Recently, beginning with ViT (Dosovitskiy et al., 2020; Caron et al., 2021; Oquab et al., 2023; Liu et al., 2021), Transformer-based network achieves revolutionary performance on the image classification challenge. To address large variations in the scale of visual entities and the high resolution of pixels in images, Swin (Liu et al., 2021) proposes a hierarchical Transformer whose representation is computed with shifted windows. On the other hand, self-supervised efforts represented by DINO (Caron et al., 2021) and DINOv2 (Oquab et al., 2023) have also been devoted to developing effective image encoder. In this paper, we propose to utilize the pre-trained image encoder as an implicit domain classifier to encode the reference images from different domains into the distinct embedding subspace (see Appendix A.1).

**Discriminator-Free Domain Adaptation.** Style-NADA (Gal et al., 2021) proposes text-driven DA and presents a discriminator-free method which utilizes CLIP (Radford et al., 2021) model to produce the CLIP-space direction. Although the method can also be applied for image-driven domain adaptation, it suffers from the semantic bias stemming from the CLIP encoder and is not guaranteed to converge to the exact realization of the style such as the *sketch* domain. Besides, the directional loss with CLIP cannot handle small attribute edits such as the *sunglasses* domain. Instead, we propose the directional subspace loss with the image encoder pre-trained on large classification dataset, which facilitates the adaptation of small attributes and significantly alleviates the bias.

## 3 METHOD

### 3.1 PROBLEM FORMULATION

Following previous works (Ojha et al., 2021; Xiao et al., 2022), we start with a pre-trained Style-GAN2 (Karras et al., 2019) generator $G_{\mathcal{S}}$ that maps from noise $z$ to images in a source domain $\mathcal{S}$, and a set of $N$ target domains $\mathcal{T}_i, i \in \{1, ..., N\}$. *Domain adaptation* (DA) finetunes $G_{\mathcal{S}}$ with few-shot

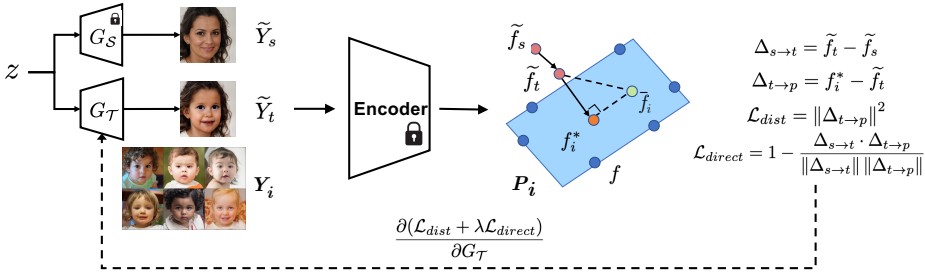

Figure 2: The diagram of our discriminator-free framework with the directional subspace loss $\mathcal{L}_{dist}$ and $\mathcal{L}_{direct}$. $\boldsymbol{Y}_i$ is the reference images from the $i$-th domain. $\widetilde{Y}_s$ and $\widetilde{Y}_t$ are images generated by frozen source generator and training target generator. The frozen image encoder extracts features $f$ to constitute the subspace $\boldsymbol{P}_i$. After that, we project the target embedding $\widetilde{f}_t$ onto the subspace $\boldsymbol{P}_i$ to obtain $f_i^*$. Then we minimize the distance between $\widetilde{f}_t$ and $f_i^*$. Simultaneously, we minimize the angle between $\widetilde{f}_t - \widetilde{f}_s$ and $f_i^* - \widetilde{f}_t$ to guarantee that $\widetilde{f}_t$ moves along the perpendicular from $\widetilde{f}_s$ to $\boldsymbol{P}_i$.

images $\boldsymbol{Y}_i$ from each target domain $\mathcal{T}_i$ to yield a generator $G_{\mathcal{T}_i}$. The adapted $G_{\mathcal{T}_i}$ can generate images $\widetilde{Y}_i$ similar to domain $\mathcal{T}_i$. Differently, *hybrid domain adaptation* (HDA) aims to acquire a generator $G_{\mathcal{T}}$ that models a hybrid domain $\mathcal{T} = \cup_{i=1}^N \mathcal{T}_i$ and generate images $\widetilde{Y}$ with integrated attributes.

## 3.2 DISCRIMINATOR-FREE DOMAIN ADAPTATION

Conventional DA methods commonly incorporate a discriminator to differentiate whether the generated images pertain to the target domain. However, we only have images sampled from individual domains and lack real images of the hybrid domain, which poses challenges in the development of discriminator-based methods. Moreover, owing to the scarcity of reference images (Ojha et al., 2021), the discriminator inevitably exhibits a tendency to overfit to easily learned attributes of certain target domains. Consequently, this leads the generator to ignore other target domains' attributes within the context of HDA.

To address the issues, we propose a discriminator-free framework for HDA. Specifically, we utilize the pre-trained image encoder $E$, such as Swin (Liu et al., 2021) and Dinov2 (Oquab et al., 2023) in image classification task, to encode the few-shot reference images $\boldsymbol{Y}_i$ from $\mathcal{T}_i$ into features $\boldsymbol{f}_i$:

$$\boldsymbol{f}_i = E(\boldsymbol{Y}_i), \quad \overline{f}_i = \frac{1}{|\boldsymbol{f}_i|} \sum_{f \in \boldsymbol{f}_i} f, \tag{1}$$

where $\overline{f}_i$ is the average mean of $\boldsymbol{f}_i$ and $|\boldsymbol{f}_i|$ is the cardinality of $\boldsymbol{f}_i$ that equals to the number of images in $\mathcal{T}_i$. As shown in Fig. 2, the features $f$ in $\boldsymbol{f}_i$ constitute a separable embedding subspace $\boldsymbol{P}_i$ (see Appendix A.1).

## 3.3 DIRECTIONAL SUBSPACE LOSS

Then we introduce a directional subspace loss for the adaptation. Technically, we encode the images generated by the frozen source generator $G_{\mathcal{S}}$ and the adapted generator $G_{\mathcal{T}}$ with the same noise $z$.

$$\widetilde{Y}_s = G_{\mathcal{S}}(z), \quad \widetilde{Y}_t = G_{\mathcal{T}}(z)$$
$$\widetilde{f}_s = E(\widetilde{Y}_s), \quad \widetilde{f}_t = E(\widetilde{Y}_t). \tag{2}$$

Our objective is for generated images to progressively approach the target subspace. Inspired by (Simon et al., 2020), we propose the distance loss from the generated images to the target subspace. Concretely, we first project the embedding $\widetilde{f}_t$ onto the subspace $\boldsymbol{P}_i$ to get the projected point $f_i^*$:

$$f_i^* = \boldsymbol{M}_i \boldsymbol{M}_i^T (\widetilde{f}_t - \overline{f}_i) + \overline{f}_i, \tag{3}$$

where $\boldsymbol{M}_i$ is the projection matrix computed from SVD factorization of all elements in $\boldsymbol{f}_i$. We then minimize the distance between $\widetilde{f}_t$ and subspace $\boldsymbol{P}_i$ spanned by $\boldsymbol{f}_i$:

$$\mathcal{L}_{dist} = dist(\widetilde{f}_t, \boldsymbol{P}_i) = \left\| f_i^* - \widetilde{f}_t \right\|^2. \tag{4}$$

However, solely employing $\mathcal{L}_{dist}$ can still result in model collapse, *i.e.*, distinct generated images may project onto the same point on $\boldsymbol{P}_i$ which compromises cross-domain consistency. To preserve more characteristics from $\widetilde{Y}_s$, we propose to enforce the collinearity between $\widetilde{f}_s$, $\widetilde{f}_t$, and $f_i^*$ such that $f_i^*$ becomes the closest point to $\widetilde{f}_s$ on the subspace $\boldsymbol{P}_i$. To this end, we minimize the angle between $\widetilde{f}_t - \widetilde{f}_s$ and $f_i^* - \widetilde{f}_t$ to guarantee that $\widetilde{f}_t$ moves along the perpendicular from $\widetilde{f}_s$ to the subspace $\boldsymbol{P}_i$:

$$\Delta_{s\to t} = \widetilde{f}_t - \widetilde{f}_s, \quad \Delta_{t\to p} = f_i^* - \widetilde{f}_t,$$
$$\mathcal{L}_{direct} = 1 - \frac{\Delta_{s\to t} \cdot \Delta_{t\to p}}{\|\Delta_{s\to t}\| \|\Delta_{t\to p}\|}, \tag{5}$$

Besides, to alleviate the biases introduced by individual image encoder, we employ the ensemble method that exploits several image encoders during the adaptation:

$$\mathcal{L}_{\text{DA}} = \mathbb{E}_{z\sim p(z)} \sum_{E\in\boldsymbol{E}} (\mathcal{L}_{dist} + \lambda\mathcal{L}_{direct}), \tag{6}$$

where $\boldsymbol{E}$ is the set of pre-trained image encoders and $\lambda$ is the balancing factor.

### 3.4 Hybrid Domain Adaptation

Naturally, we can extend the directional subspace loss for HDA, since DA is a special form of HDA. For the distance loss, we reduce the distances between $\widetilde{f}_t$ and all target subspaces to ensure that the adapted generator owns their attributes:

$$\mathcal{L}'_{dist} = \sum_{i=1}^{N} \alpha_i \left\| f_i^* - \widetilde{f}_t \right\|^2, \tag{7}$$

where $\alpha_i$ is pre-defined domain coefficient for the $i$-th domain.

On the other hand, we also extend the direction loss for HDA to preserve the characteristics of the source domain $\widetilde{f}_s$. Specifically, we minimize the angle between $\widetilde{f}_t - \widetilde{f}_s$ and the weighted sum of $f_i^* - \widetilde{f}_t$ (the perpendicular from $\widetilde{f}_t$ to the $i$-th subspace):

$$\Delta'_{t\to p} = \sum_{i=1}^{N} \alpha_i (f_i^* - \widetilde{f}_t),$$
$$\mathcal{L}'_{direct} = 1 - \frac{\Delta_{s\to t} \cdot \Delta'_{t\to p}}{\|\Delta_{s\to t}\|\|\Delta'_{t\to p}\|}. \tag{8}$$

Overall, our final objective for HDA is

$$\mathcal{L}_{\text{HDA}} = \mathbb{E}_{z\sim p(z)} \sum_{E\in\boldsymbol{E}} (\mathcal{L}'_{dist} + \lambda\mathcal{L}'_{direct}), \tag{9}$$

## 4 Experiments

### 4.1 Datasets and Metrics

**Datasets:** Following previous literature, we consider Flickr-Faces-HQ (FFHQ) (Karras et al., 2019) as one of the source domains and adapt to the combination of the following individual target domains (a) FFHQ-*baby*, (b) FFHQ-*sunglasses*, (c) *sketch*, and (d) FFHQ-*smile*. As in previous works, all our experiments use 10 randomly sampled targets for each domain. Unless stated otherwise, we operate on $256 \times 256$ resolution images for both the source and target domains.

**Metrics:** A key difference between the proposed HDA task against previous DA (Ojha et al., 2021; Mondal et al.; Nitzan et al., 2023) is that there are no real images in the hybrid target domains. In terms of the quantitative evaluation, we focus on these evaluation metrics in our experiments: (1) CLIP-Score (Nitzan et al., 2023; Radford et al., 2021) measures the compatibility of image-text pairs, which can be thought of as the semantic similarity with the target domain. (2) The Inception Score (IS) is used to assess the fidelity of images, which is practical in the few shot setting (Xiao et al., 2022). (3) Identify similarity (ID) measures cross-domain consistency where the adapted images retain consistency with their corresponding source images in domain-invariant aspects like pose and identity, following (Zhang et al., 2022; Wu et al., 2023).

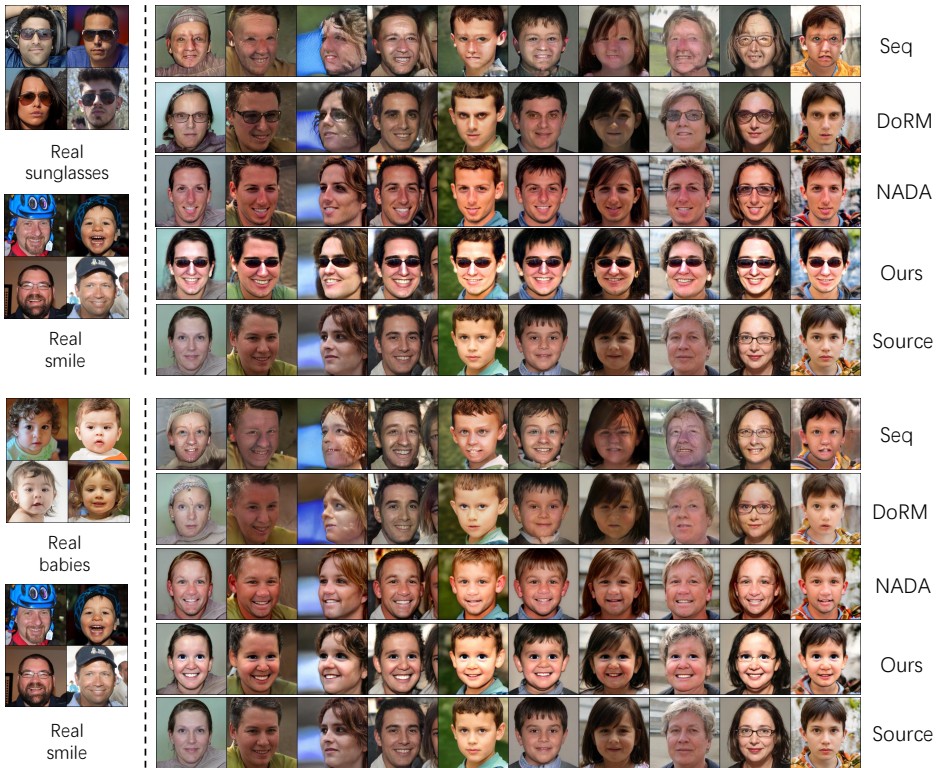

Figure 3: Qualitative results on 10-shot HDA of *smile-sunglasses* and *baby-smile*.

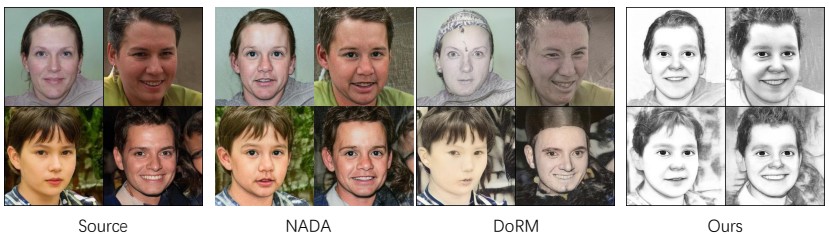

Figure 4: Qualitative results on 3 domain 10-shot HDA of *sketch-smile-baby*.

## 4.2 METHODOLOGY AND BASELINES

**Methodology:** Following previous domain adaptation (DA) works (Ojha et al., 2021; Xiao et al., 2022; Mondal et al.; Zhao et al., 2022b), we apply our method on the StyleGAN2 (Karras et al., 2020) architecture. To show the flexibility of our method, we conduct the experiments on both DA and HDA. As described in Section 3.2, we utilize Swin (Liu et al., 2021) and Dinov2 (Oquab et al., 2023) as the image encoder, which are pre-trained on ImageNet 22k dataset (Deng et al., 2009).

**Baselines:** To demonstrate the effectiveness of our method, we compare our method against the following baselines that could potentially achieve hybrid domain: (1) Sequential: sequentially adapted to individual target domains to achieve hybrid domain; (2) DoRM: Domain Remodulation (Wu et al., 2023) for hybrid domain as discussed in Section 2; (3) NADA: Following NADA (Gal et al., 2021), we interpolate the model parameters in different domains to achieve hybrid domain. Additionally for DA, we compare ours with CDC (Ojha et al., 2021), DoRM (Wu et al., 2023), RSSA (Xiao et al., 2022), AdAM (Zhao et al., 2022a), and NADA (Gal et al., 2021).

## 4.3 RESULTS ON HYBRID DOMAIN ADAPTATION

**Qualitative Results:** Fig. 3 shows the qualitative result of the baseline and our method on HDA, all of which start from the same source domain FFHQ (Karras et al., 2019) to the combinations of individual

| Method | smile-sunglass | | | baby-smile | | | baby-sketch | | |
|--------|--------|--------|--------|--------|--------|--------|--------|--------|--------|
| | CS (↑) | IS (↑) | ID (↑) | CS (↑) | IS (↑) | ID (↑) | CS (↑) | IS (↑) | ID (↑) |
| Seq | 17.92 | 1.82 | 0.261 | 17.36 | 1.7 | 0.33 | 22.69 | 2.28 | 0.198 |
| DoRM | 18.99 | 1.94 | 0.312 | 16.38 | 2.12 | 0.28 | 17.46 | 2.69 | 0.21 |
| NADA | 20.23 | 1.34 | 0.27 | 20.45 | 1.68 | **0.34** | 22.06 | 2.61 | 0.18 |
| Ours | **25.65** | **2.4** | **0.373** | **23.08** | **2.94** | **0.34** | **24.91** | **2.97** | **0.223** |

Table 1: Quantitative evaluations on 10-shot HDA including CLIP-Score (CS), IS, and ID discussed in Section 4.1. Note that ↑ indicates higher is better.

| Method | Inter. | Dis. free | 2-domain | | 5-domain | | 10-domain | |
|--------|--------|-----------|--------|--------|--------|--------|--------|--------|
| | | | size (↓) | time (↓) | size (↓) | time (↓) | size (↓) | time (↓) |
| Seq | | | 24M | 240min | 24M | 600min | 24M | 1200min |
| DoRM | ✓ | | 30M | 240min | 54M | 600min | 84M | 1200min |
| NADA | ✓ | ✓ | 48M | 6min | 120M | 15min | 240M | 30min |
| Ours | | ✓ | **24M** | **3min** | **24M** | **3min** | **24M** | **3min** |

Table 2: Comparison of model size and training time. Note that *Inter.* and *Dis. free* indicate the inference time interpolation and discriminator-free methods.

*baby*, *sketch*, *smile* and *sunglasses* domains, respectively. For naive Sequential learning, we find that it suffers from catastrophic forgetting (McCloskey & Cohen, 1989) and tends to generate images of the last adapted domain, which fails to generate desired images (*e.g.*, *baby with the style of sketch*). DoRM (Wu et al., 2023) are unstable to generate images of the hybrid domain (*e.g.*, the images of *smile-sunglasses* lack the attribute of *smile* and lose the characteristics of the source domain), since the simple interpolation of latent codes at inference time is not enough to semantically align multiple domains. NADA could somehow generate images with integrated attributes, but it exhibits significant bias stemming from the CLIP model. For instance, the sketch domain deviates stylistically from the reference images, and it fails to learn the sunglasses domain. In contrast, our method exhibits desirable performance and shows its stable capability to generate hybrid target images and preserve the characteristics of the source domain. Furthermore, we also conduct experiments on the hybrid of three target domains *sketch-smile-baby* in Fig. 4 where our generated images blend the domain-specific attributes and preserve the primary characteristics of the source domain, which surpass all the baseline methods. More results for HDA are included in Appendix.

**Quantitative Results:** To quantify the quality and diversity of the generated images, we evaluate all methods with CLIP-Score, IS, and ID respectively. As shown in Table 1, our method achieves best scores for all metrics. Especially for CLIP-Score, our method significantly outperforms other methods, indicating that generated images with our method effectively integrate the multiple attributes from distinct domains. Furthermore, our method achieves better IS and ID, indicating that generated images has higher fidelity and preserve more characteristics of the source domain.

Additionally, we calculate the model size and training time to verify the efficiency of our method as shown in Table 2. Compared with the discriminator-based baselines, our discriminator-free method obviates adversarial training, thereby saving substantial training time. Furthermore, compared with interpolation-based approaches at inference time, our method saves multiple times the model size and training time, completing the adaptation in just three minutes.

## 4.4 RESULTS ON SINGLE DOMAIN ADAPTATION

**Qualitative Results:** To show the flexibility of our proposed method, we also conduct the experiments on DA, as shown in Fig. 5. Similar to the performance on HDA, our method surpasses all baseline approaches. Specifically, previous methods exhibit severe overfitting to the few-shot reference images, inadequately preserving source domain attributes, while also demonstrating relatively inferior image fidelity. While NADA demonstrates cross-domain consistency to some extent, it exhibits significant

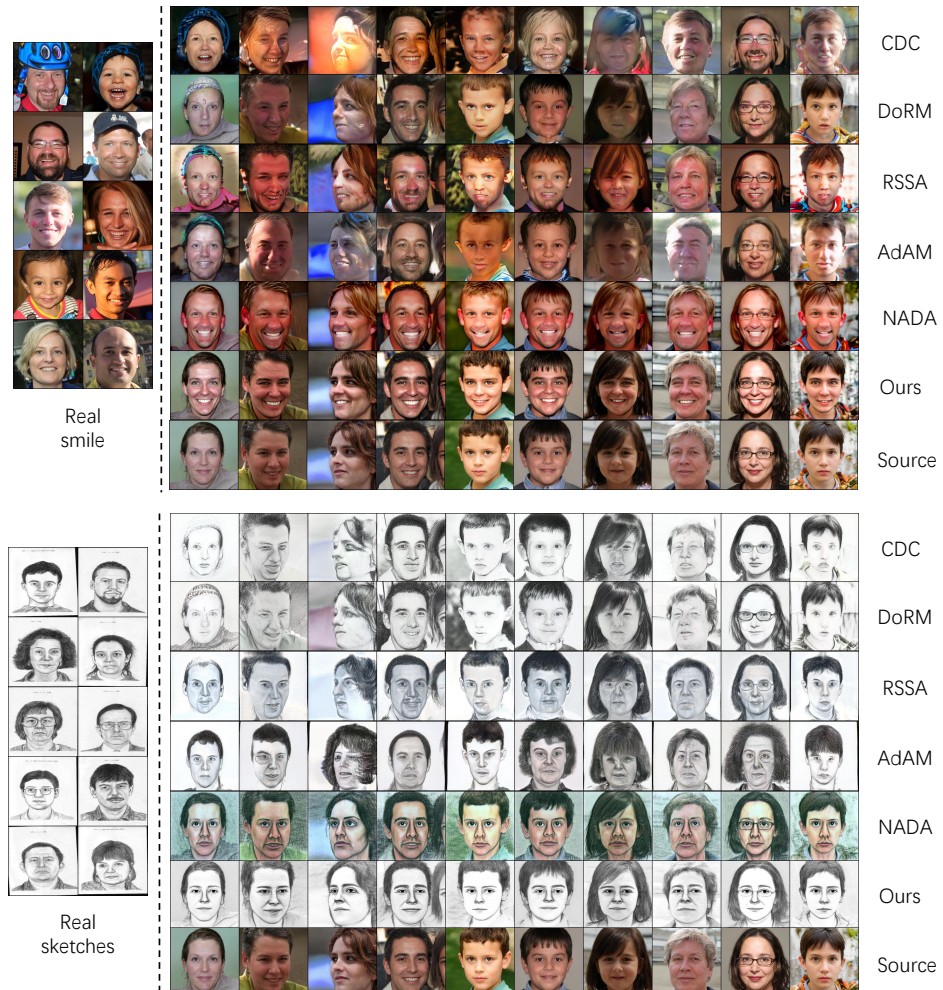

Figure 5: Qualitative results on 10-shot DA of *smile* and *sketch*.

| Method | *baby* | | | *smile* | | | *sunglasses* | | | *sketch* | | |
|--------|--------|--------|--------|--------|--------|--------|--------|--------|--------|--------|--------|--------|
| | CS (↑) | IS (↑) | ID (↑) | CS (↑) | IS (↑) | ID (↑) | CS (↑) | IS (↑) | ID (↑) | CS (↑) | IS (↑) | ID (↑) |
| CDC | 17.84 | 2.22 | 0.326 | 18.93 | 1.22 | 0.479 | 22.19 | 2.33 | 0.318 | 20.83 | 2.12 | 0.214 |
| DoRM | 17.10 | 2.30 | 0.335 | 17.10 | 2.18 | 0.490 | 22.26 | 2.59 | 0.389 | 19.42 | 2.06 | 0.365 |
| RSSA | 18.42 | 2.57 | 0.238 | 18.18 | 2.20 | 0.410 | 21.63 | 2.52 | 0.256 | 21.40 | 1.66 | 0.231 |
| AdAM | 21.18 | 2.12 | 0.210 | 19.79 | 1.61 | 0.280 | 22.50 | 2.50 | 0.197 | 21.18 | 2.22 | 0.109 |
| NADA | 21.52 | 3.36 | 0.182 | 19.95 | 1.18 | 0.305 | 20.95 | 1.51 | 0.233 | 20.9 | 1.86 | 0.335 |
| Ours | **21.92** | **3.71** | **0.353** | **20.38** | **2.23** | **0.587** | **24.13** | **2.67** | **0.414** | **22.83** | **2.39** | **0.417** |

Table 3: Quantitative evaluations on 10-shot DA.

bias stemming from the CLIP encoder. In contrast, our method is capable of accurately acquiring the visual attributes of target domain while retaining strong fidelity and cross-domain consistency.

**Quantitative Results:** To demonstrate the effectiveness of our method on DA, we evaluate all the methods with CLIP-Score, IS, and ID respectively. As reported in Table 3, our method surpasses all the baselines. Concretely, our method achieves higher semantic similarity with the target domain in CLIP-Score. Furthermore, it outperforms priors on IS and ID, indicating that our generated images have better fidelity and cross-domain consistency.

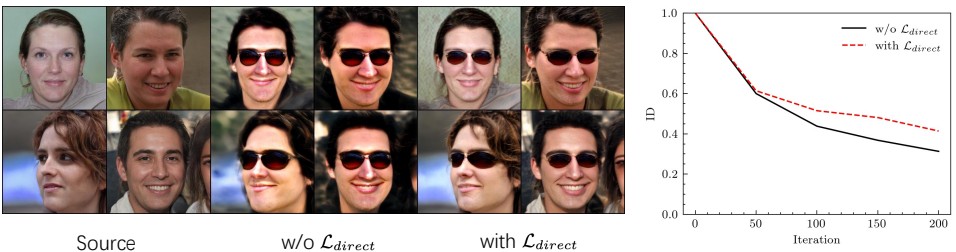

Figure 6: Qualitative ablation for $\mathcal{L}_{direct}$ on *sunglasses*.

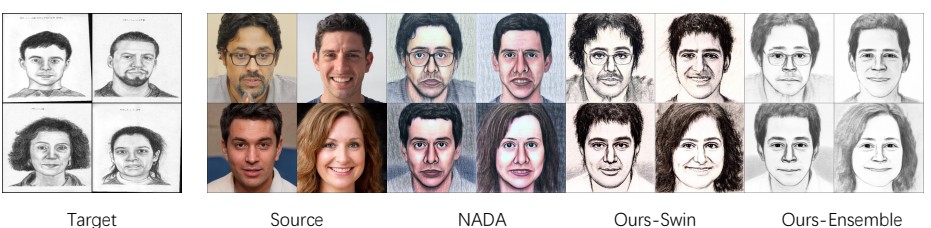

Figure 7: Qualitative ablation for the ensemble on *sketch*.

### 4.5 ABLATION STUDY

First, we conduct the ablation study to verify the effectiveness of our proposed directional subspace loss. As shown in Fig. 6, $\mathcal{L}_{direct}$ greatly improves the cross-domain consistency (*e.g.*, clothes and hair). Consistently, we observe that the adaptation with our directional loss maintains higher ID scores throughout training. Quantitatively, we report the ablation for $\mathcal{L}_{dist}$ and $\mathcal{L}_{direct}$ in Table 4. It can be clearly observed that

| $\mathcal{L}_{dist}$ | $\mathcal{L}_{direct}$ | sunglass | | baby | |
|:---:|:---:|:---:|:---:|:---:|:---:|
| | | IS (↑) | ID (↑) | IS (↑) | ID (↑) |
| ✓ | | 1.91 | 0.249 | 2.56 | 0.178 |
| | ✓ | 2.61 | 0.372 | 3.5 | 0.304 |
| ✓ | ✓ | **2.67** | **0.414** | **3.71** | **0.353** |

Table 4: Ablation for $\mathcal{L}_{dist}$ and $\mathcal{L}_{direct}$ on 10-shot DA.

the $\mathcal{L}_{dist}$ and $\mathcal{L}_{direct}$ significantly improve image fidelity and cross-domain consistency. Besides, we report the qualitative ablation for the encoders ensemble in Fig. 7. Our method with Swin (Liu et al., 2021) significantly alleviates the bias compared with NADA (Gal et al., 2021) and the ensemble technique further help to converge to the exact realization of the style of the *sketch* domain.

## 5 CONCLUSION & LIMITATION

**Conclusion.** In this paper, we propose a new task – Hybrid Domain Adaptation (HDA), which aims to adapt a pre-trained generator to the hybrid target domain to generate images with integrated attributes. To achieve this, we introduce a novel discriminator-free framework, which utilizes the pre-trained image encoder to encode the reference images from different domains into separate embedding subspaces. Furthermore, we introduce a directional subspace loss to guide the adaptation and retain cross-domain consistency. We believe that our work is an important step towards few-shot HDA since we have demonstrated that the source generator can be effectively adapted to a hybrid domain without accessing the real images from the hybrid domain.

**Limitation.** While our method realizes HDA and achieves better superior metrics compared with state-of-the-art DA methods, it also has the limitation. We observe that for hybrid domain adaptation, the weights of different domain in our loss are not entirely equivalent to the proportionality of visual characteristics, which needs empirical adjustment. Nevertheless, we believe that some solutions could be integrated into our HDA pipeline to facilitate the alignment in the future.

## 6 ACKNOWLEDGEMENT

This work was supported in part by The National Nature Science Foundation of China (Grant No: 62303406), in part by Yongjiang Talent Introduction Programme (Grant No: 2023A-194-G).

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

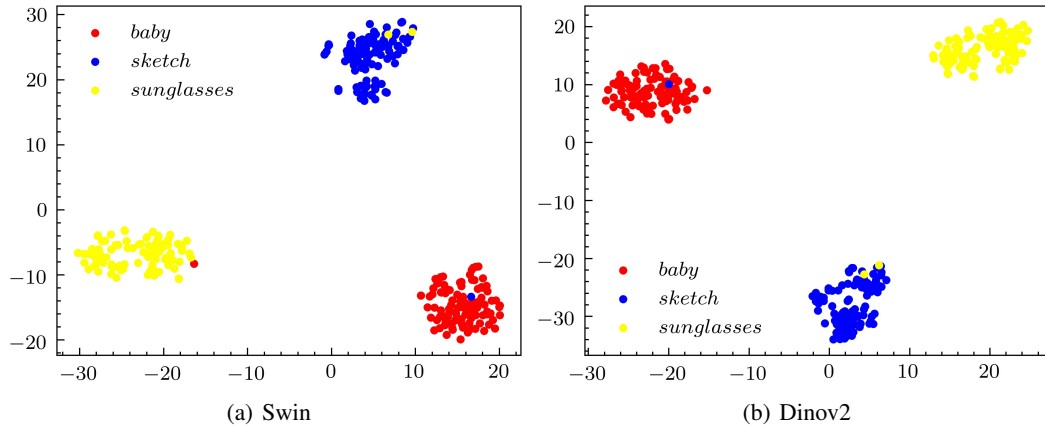

Figure 8: T-SNE visualization of the embedding space encoded by Swin (Liu et al., 2021) and Dinov2 (Oquab et al., 2023) for images from *baby*, *sketch*, and *sunglasses* domain.

| Intra-LPIPS (↑) | *smile* | *sketch* | *sunglasses* |
|---|---|---|---|
| NADA | 0.529 | 0.491 | 0.535 |
| CDC | 0.616 | 0.453 | 0.562 |
| RSSA | 0.625 | 0.480 | 0.563 |
| AdAM | 0.615 | 0.495 | 0.598 |
| Ours | **0.642** | **0.506** | **0.615** |

Table 5: Intra-cluster pairwise LPIPS distance on 10-shot DA.

# A APPENDIX

## A.1 EMBEDDING SPACE ANALYSIS

Since the image encoder in image classification typically projects the images into a well-separable embedding space, such as Swin (Liu et al., 2021) and Dinov2 (Oquab et al., 2023), we can employ the separable subspaces to achieve HDA. To validate that, we visualize the embeddings of real images from multiple domains by t-SNE (Van der Maaten & Hinton, 2008). As shown in Fig. 8, distinct subspaces from different domains manifest separation, which facilitates us to adapt the generator to the target subspace corresponding to the target domain.

## A.2 INTRA-LPIPS

Following CDC (Ojha et al., 2021), we utilize Intra-cluster pairwise LPIPS distance (Zhang et al., 2018) to evaluate the diversity of generated images for single domain adaptation. Specifically, we use 1000 generated images and 10 real images from the target domain to compute Intra-LPIPS. As shown in Table 5, our method achieves higher Intra-LPIPS distance than baseline methods, indicating more distinct images being generated.

## A.3 MORE QUALITATIVE RESULTS

As the complement of qualitative results in the main paper, we conduct the experiments on more target domains. We present the results in Fig. 9 and Fig. 10 for DA and HDA respectively. It can be observed that the generated images generated by our methods integrate the attributes, which significantly surpasses the baseline methods.

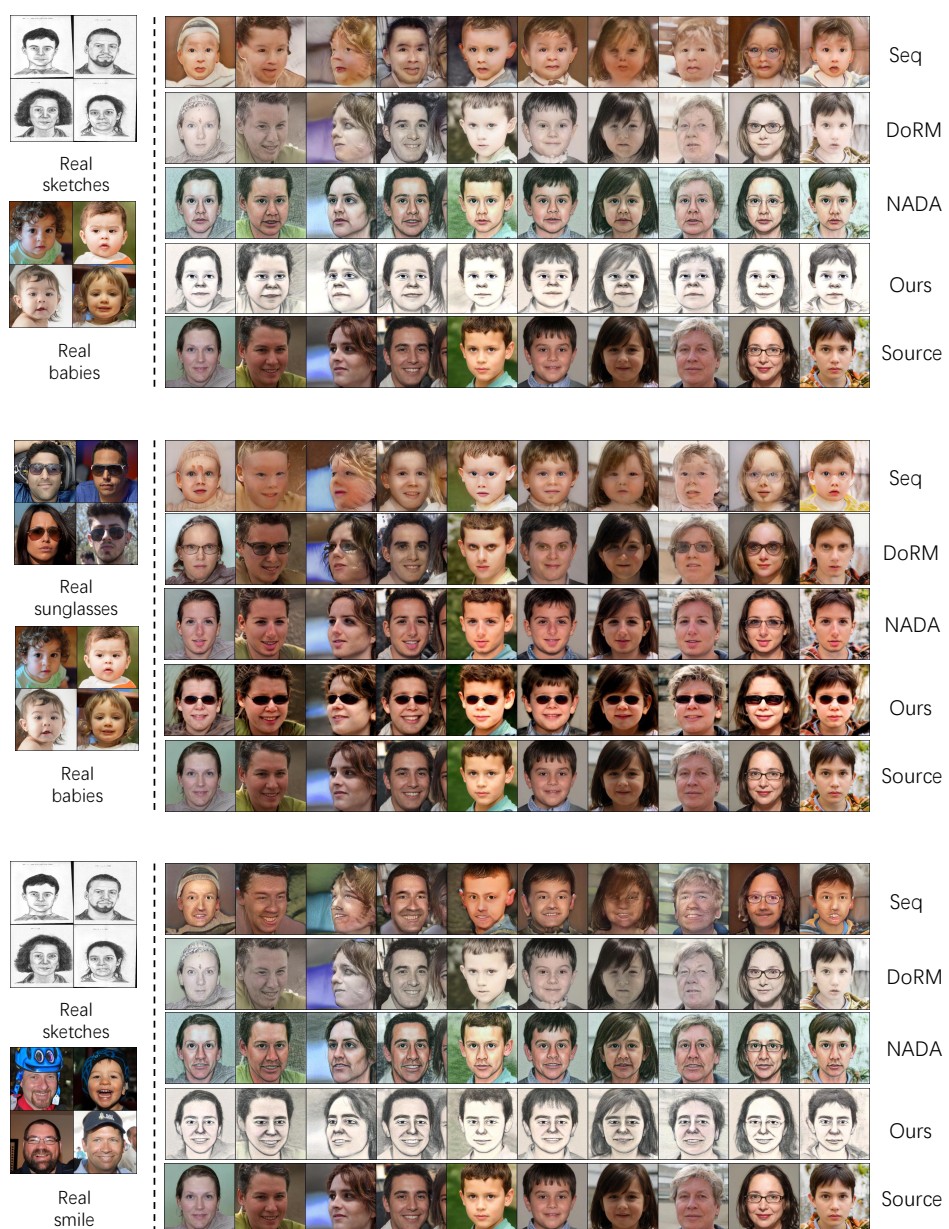

Figure 9: Qualitative results on 10-shot HDA.

## A.4 IMPLEMENT DETAILS

As depicted in Eq. (7) and Eq. (8), we use pre-defined domain coefficient to modulate the attributes from multiple domains. For most experiments on two domains, we use $\alpha_i = 0.5$ except for *baby-sunglasses* with $\alpha_{baby} = 0.3$ and $\alpha_{sunglasses} = 0.7$. For the experiment of three domains, we set $\alpha_{baby} = 0.7$, $\alpha_{sketch} = 0.4$, and $\alpha_{smile} = 0.3$.

Following the setting of previous DA methods, we utilize the batch size of 4 and a training session typically requires 300 iterations in roughly 3 minutes on a single NVIDIA TITAN GPU. Besides, we set the balancing factor $\lambda$ as 1 in Eq. (6) and Eq. (9).

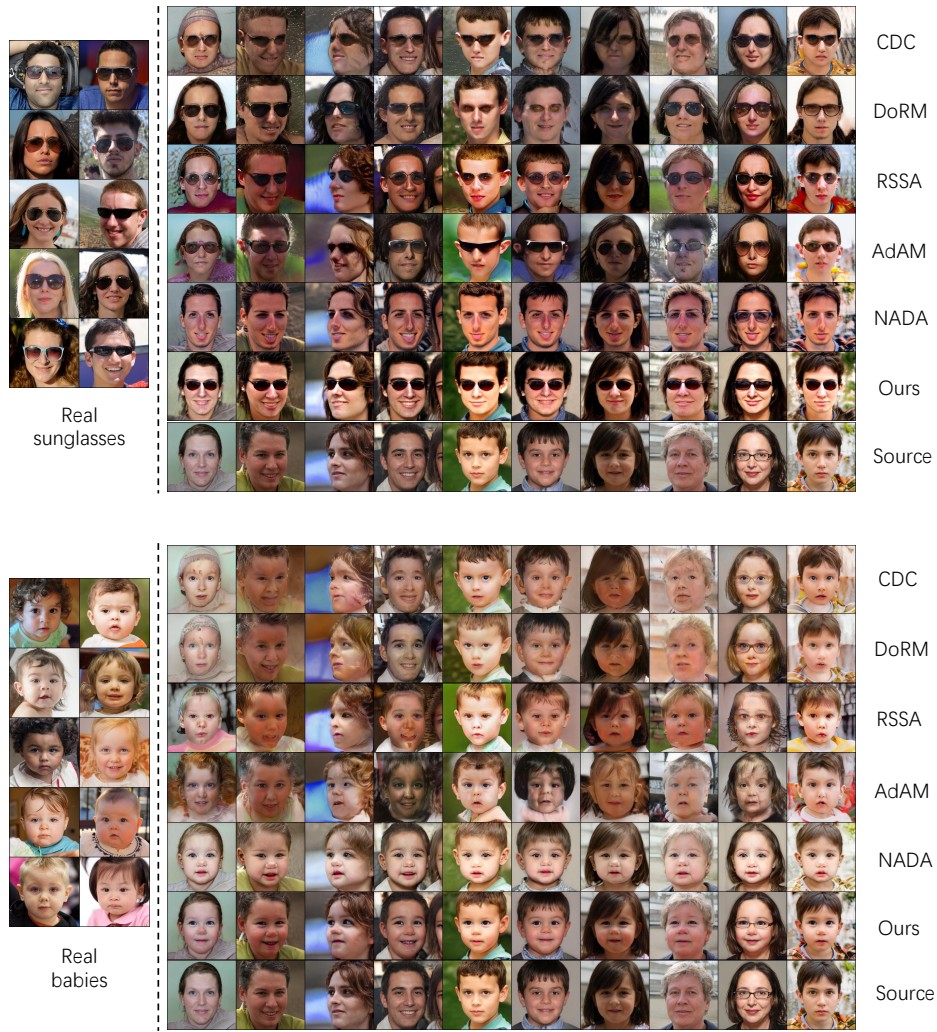

Figure 10: Qualitative results on 10-shot DA of *sunglasses* and *baby*.

| Domain | Prompt |
|---|---|
| *baby* | "face of a baby" |
| *sketch* | "face of a person with the style of sketch" |
| *sunglasses* | "face of a person with sunglasses" |
| *smile* | "face of a person with smile" |
| *smile-sunglasses* | "face of a person with smile and sunglasses" |
| *baby-smile* | "face of a baby with smile" |
| *baby-sketch* | "face of a baby with the style of sketch" |

Table 6: Prompts of different domains for CLIP-Score.

## A.5 PROMPTS FOR CLIP-SCORE

To measure the semantic similarity with the target domain, we adopt CLIP-Score as the evaluation metric in Section 4.1. As shown in Table 6, we present the prompts corresponding to different target domains.

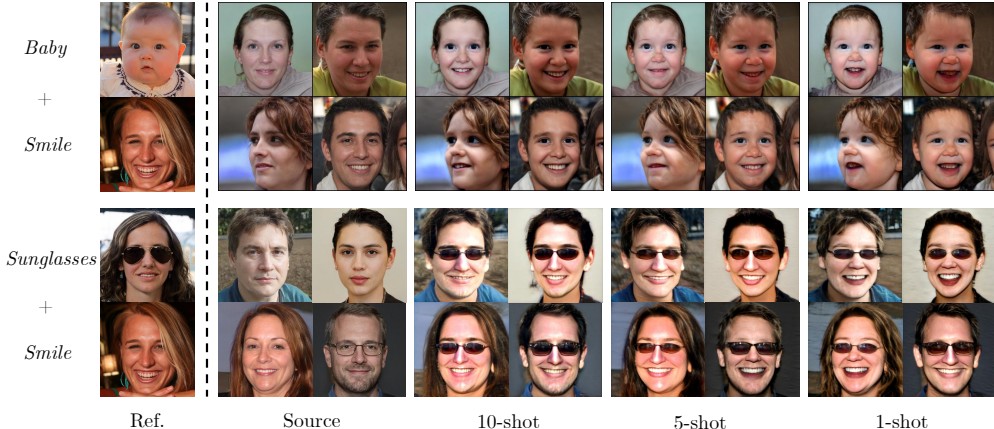

Figure 11: Impact of different shots for our method in HDA. Results of 5-shot are close to those of 10-shot. Compared with them, the results of 1-shot exhibit relatively lower cross-domain consistency.

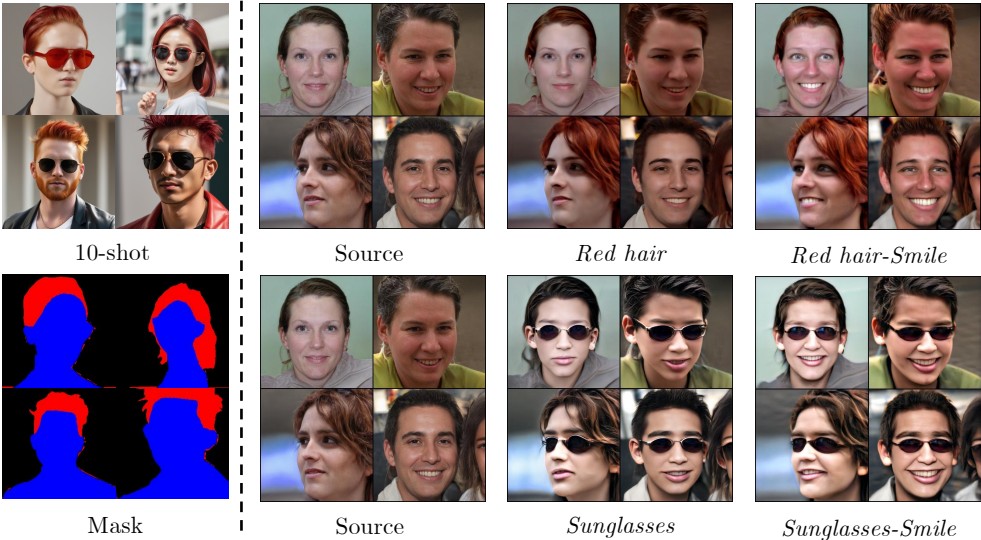

Figure 12: Conditional generation of our method in DA and HDA. We adapt the generator with masked images for *red hair* and *sunglasses* respectively. The red masks represent *red hair*, while the blue masks represent faces wearing *sunglasses*.

A.6 RESULTS USING DIFFERENT NUMBERS OF SHOTS

We perform experiments on lower shots (5-shot and 1-shot) as suggested using various datasets. As shown in Fig. 11, the results of 5-shot are close to those of 10-shot, which integrates the attributes and maintains the consistency with source domain. Although multiple attributes have been learned, the results of 1-shot exhibit relatively lower cross-domain consistency compared with 10-shot and 5-shot. This is because when there is only one reference image per domain, the subspace in our directional subspace loss degenerates into a single point (see Fig. 2). Then distinct generated images corresponding to different noise tend to converge towards the same image in CLIP's embedding space, which comprises cross-domain consistency as depicted in Section 3.3.

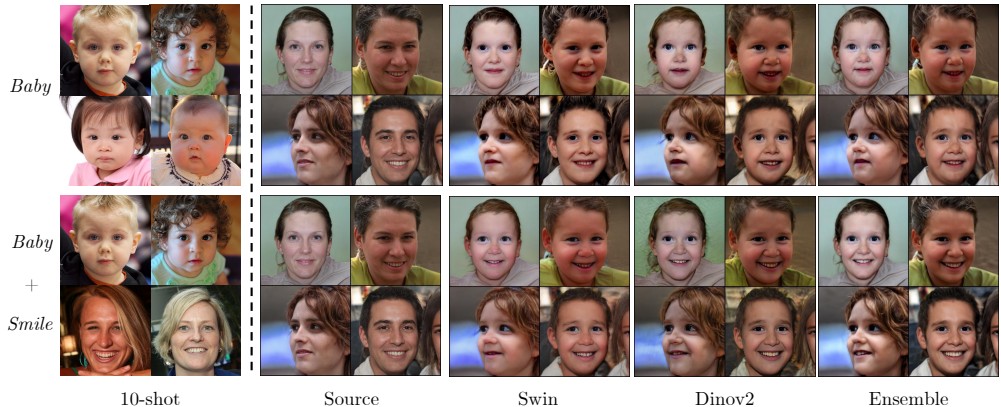

Figure 13: Effect of different pre-trained image encoders in Fig. 2, *e.g.*, Swin, Dinov2, and their ensemble.

## A.7   RESULTS OF CONDITIONAL GENERATION

As shown in Fig. 12, we conduct the experiments on conditional generation. Specifically, we collect 10-shot images with *red hair* and *sunglasses*. Then we use masks to separate these attributes and adapt the generator with masked images for *red hair* and *sunglasses* respectively. We can observe that the generated images possess the corresponding attribute for both single DA and hybrid DA. Simultaneously, these images also maintain consistency with source domain.

## A.8   EFFECT OF DIFFERENT PRE-TRAINED IMAGE ENCODERS

As shown in Fig. 13, we conduct experiments on pre-trained Swin and Dinov2 to explore the impact of different image encoders on generated images. Our method is agnostic to different pre-trained image encoders. Although they exhibit slight stylistic differences, these are due to their different approaches to extract features into separated subspaces, as depicted in Fig. 8. To converge to the exact realization of the target domain, our method employs the ensemble technique that exploits both Swin and Dinov2. As shown in the figure, the results closely resembles the attributes of the target domain while maintaining the best consistency with source domain.

## A.9   TRAVERSAL FOR DOMAIN COEFFICIENT $\alpha$

In our method for hybrid domain adaptation, this parameter controls the composition ratio of the attribute from each domain. As depicted in Appendix A.4, we use $\alpha_i = 0.5$ for most experiments without the need for complex and intricate adjustments. To further explore the sensitivity, we conduct the study for simple traversal of $\alpha_i$. As shown in Fig. 14, the attributes of generated images transit smoothly between domains. Our method produces the similar attribute blending effect when $\alpha_i \in \{0.4, 0.5, 0.6\}$.

## A.10   THE RESULTS OF MORE DOMAINS

Consistent with prior work for few-shot generative domain adaptation (like CDC (Ojha et al., 2021), RSSA (Xiao et al., 2022) and DoRM (Wu et al., 2023)), our experimental data encompasses both global style (*sketch* and *baby*) and local attributes (*smile* and *sunglasses*). The combination of these domains demonstrates the effectiveness of our method since they encompass all types of generative domain adaptation.

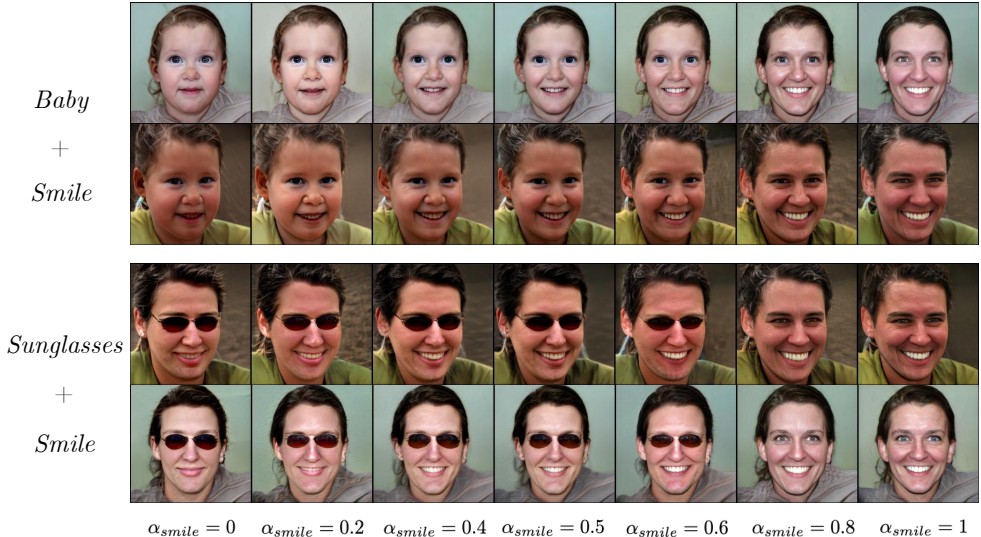

$\alpha_{smile} = 0 \quad \alpha_{smile} = 0.2 \quad \alpha_{smile} = 0.4 \quad \alpha_{smile} = 0.5 \quad \alpha_{smile} = 0.6 \quad \alpha_{smile} = 0.8 \quad \alpha_{smile} = 1$

Figure 14: The study of simple traversal on $\alpha_i$ for hybrid domain adaptation. The attributes of generated images transit smoothly between two domains. Our method produces the similar attribute blending effect when $\alpha_i \in \{0.4, 0.5, 0.6\}$. Note that $\alpha_{baby}$ and $\alpha_{sunglasses}$ are $1 - \alpha_{smile}$.

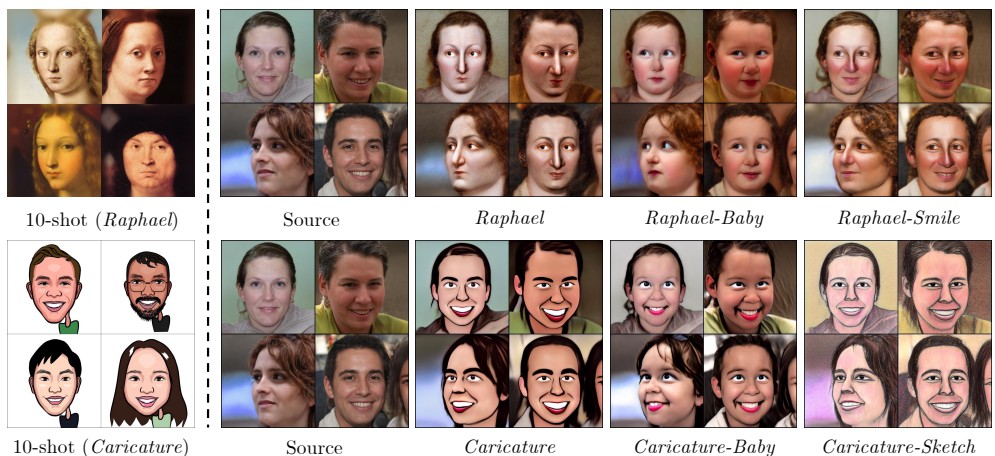

Figure 15: Single and hybrid domain adaptation on *Raphael* and *Caricature*.

To provide more comprehensive evidence of validity, we conduct additional experiments on *Raphael* and *Caricature* for both single and hybrid domain adaptation. As shown in Fig. 15, the results integrate the characteristics from multiple target domains and maintains robust consistency with source domain, which further demonstrates the effectiveness of our method.

## A.11 THE GENERALIZABILITY TO OTHER DOMAINS

To verify the generalizability of our method to other domains, we conduct experiments on church domain following prior work (Ojha et al., 2021; Xiao et al., 2022; Wu et al., 2023), RSSA (Xiao et al., 2022) and DoRM (Wu et al., 2023)). We adapt the pre-trained generator from *LSUN Church* (Yu et al., 2015) to *Van Gogh's house paintings*, *haunted houses*, and the combination of them. As shown in Fig. 16, the results acquire the corresponding style and showcase the preservation of good cross-domain consistency. This aligns with the results observed in the face domain.

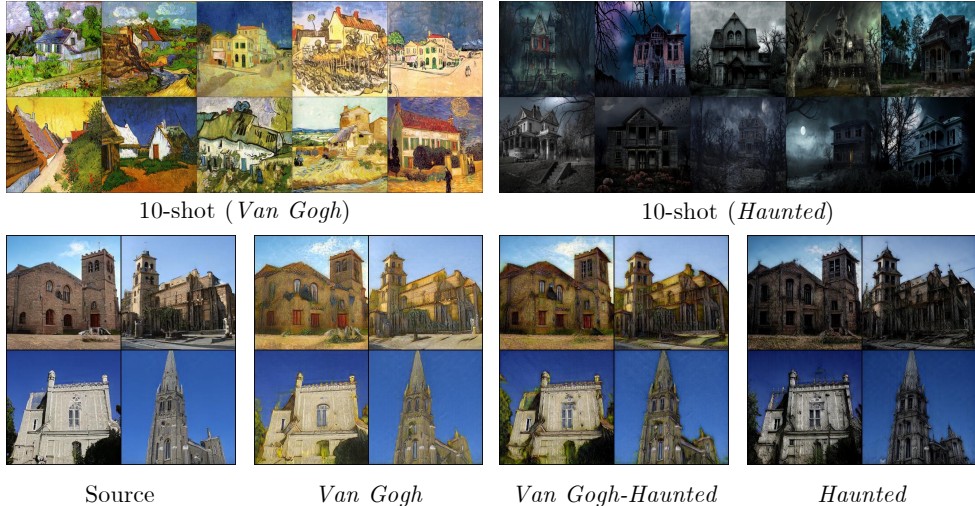

10-shot (*Van Gogh*)          10-shot (*Haunted*)

Source     *Van Gogh*     *Van Gogh-Haunted*     *Haunted*

Figure 16: Results of domain adaptation from *LSUN Church* to *Van Gogh's house paintings*, *haunted houses*, and the combination of them.

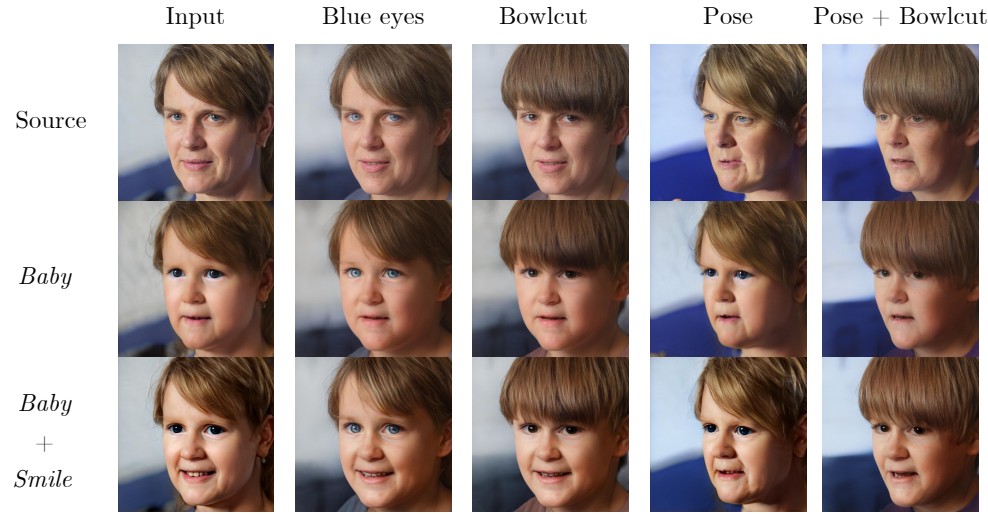

Figure 17: Editing results on the images generated by original and adapted model for both single and hybrid domain adaptation. The first line is the source images. The second and third lines are images of the adaptation.

## A.12 THE EDITABILITY BEFORE AND AFTER THE DOMAIN ADAPTATION

We conduct the editing on the images generated by original and adapted model for both single and hybrid domain adaptation. As shown in the Fig. 17, the results indicate the adapted generator maintains similar editability like pose to the original generator. This verifies that the our method effectively preserves original generator's attributes.

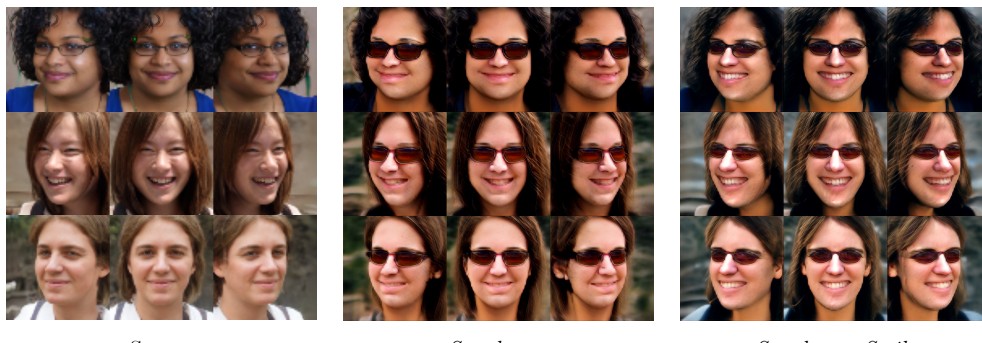

| Source | *Sunglasses* | *Sunglasses-Smile* |

Figure 18: Single and hybrid domain adaptation in 3D GAN. We adapt pre-trained EG3D (Chan et al., 2022) with our method using 10-shot training images per domain.

## A.13 SINGLE AND HYBRID DOMAIN ADAPTATION IN 3D GAN.

We conducted experiments using the popular 3D-aware image generation method, EG3D (Chan et al., 2022). Specifically, we replace the discriminator as we did in Fig. 2 for both single and hybrid domain adaptation. As shown in Fig. 18, we adapt the pre-trained generator from FFHQ to *sunglasses* and the hybrid of *sunglasses* and *smile* with 10-shot training images per domain. We can observe that the results effectively integrate the attributes and preserve the characters and poses of source domain.

## A.14 COMPARISON TO METHODS FOR DIFFUSION MODEL PERSONALIZATION

Current trend in customized text-to-image models like DreamBooth (Ruiz et al., 2023) aim to mimic the appearance of subjects in a given reference set. Similar to ours, DreamBooth fine-tunes a pre-trained generator for the personalization. However, there are two key differences between personalization and HDA.

(1) DreamBooth aims to retain the individuals from the training images, requiring similar individuals across the training set. Differently, our goal is to acquire the attributes of target domain like and preserve consistency with the source generator, ensuring that images generated from the same noise exhibit similar individuals.

(2) Dreambooth utilizes text-to-image generator which necessitates intricate and laborious adjustments of prompts to synthesize images with specific attributes. Additionally, certain attributes are challenging to accurately describe using text, such as artistic paintings. Conversely, our adapted model adeptly preserves the domain-specific attributes of reference images without the need for intricate prompt engineering.

