# OpenReview forum: "Few-shot Hybrid Domain Adaptation of Image Generator"
_ICLR.cc/2024/Conference — ICLR 2024 poster_

### Official Review · Reviewer_bxKn · 2023-10-31

**Soundness:** 3 good
**Presentation:** 3 good
**Contribution:** 3 good
**Rating:** 5
**Confidence:** 2

**Summary:**

The paper proposes a method called  Hybrid Domain Adaptation (HDA), which aims to align a GAN generator to several target domains without affecting the attributes of the source domain. As opposed to domain adaptation, this task makes sure that the attributes from the target domains do not override the attributes of the source domain. So, in essence, it is a more difficult task than conventional domain adaptation where only two domains are involved. To approach this problem, the paper proposes to use two ideas 1) use a discriminator-free framework that directly encodes different domains’ images into well-separable subspaces.  2) The paper proposes to use two types of CLIP-based losses i) distance loss to make sure the features of the source and the target domains are aligned and ii) direction loss to push the features of the target domain perpendicular to the source domain. Various qualitative results are shown throughout the paper. The paper also compares quantitatively with other domain adaptation methods.

**Strengths:**

1) The paper proposed a task more challenging than domain adaption. This task aims to adapt multiple domains in a single generator without affecting the characteristics of the original generator. To achieve this, the paper employs a discriminator-free approach and CLIP-based losses to ensure the fidelity of the outputs.

2) The paper shows results in a few shot settings and compares them with the competing domain adaptation methods. The quantitative results show that the proposed method is also faster than these methods.

3) In the context of the face domain, the paper uses to adapt sketch images which have a significant domain gap with the faces. The results show better domain adaptation compared to the competing methods. The authors also show ablation of the components used in their method.

**Weaknesses:**

1) One notable concern is the lack of diverse domain results in the paper. The primary focus of the paper revolves around attributes like age, smile, and sketch, which are maintained throughout the paper. This limited scope may not provide sufficient evidence to establish the method's overall validity.

2) Building on the previous point, the paper predominantly concentrates on face datasets, which restricts the method's evaluation. This is particularly problematic since the method itself lacks face-specific components.

3) The paper overlooks discussions and validation concerning the advantages and disadvantages of employing few-shot domain adaptation techniques with discrimination other than StyleGAN-NADA, as seen in methods such as DualStyleGAN and 3DAvatarGAN.

4) Attributes like smile and age can be manipulated using existing StyleGAN editing methods. It's challenging to discern why hybrid domain adaptation is necessary in these cases.

5) To ensure that the original generator's attributes are preserved, a useful approach could involve comparing the editability before and after the domain adaptation task. For instance, evaluating the ability to edit attributes like pose using a 2D generator. Such evaluation is lacking in the paper. This also raises the question of how the method would perform in a 3D GAN setting.

6) It is worth noting that there are other relevant papers in the field, such as "Mind the GAP," which employ similar loss mechanisms to constrain domains. Additionally, "StyleCLIP" introduces directional loss terms based on CLIP.

**Questions:**

1) Can you explain the rationale behind focusing on a limited set of attributes (age, smile, and sketch) and how this choice demonstrates the validity of the proposed method in diverse domains?

2) Given that the method mainly revolves around face datasets and lacks face-specific components, how does this impact the generalizability of the method to other domains?

3) Could you discuss and provide insights into the advantages and disadvantages of your method compared to other approaches like DualStyleGAN and 3DAvatarGAN, which utilize few-shot domain adaptation with discrimination?

4) In cases where attributes like smile and age can be manipulated using existing StyleGAN editing methods, what benefits does your hybrid domain adaptation method bring to these scenarios?

5) To determine whether the original generator's attributes are retained, could you elaborate on the results of comparing editability before and after the domain adaptation task?

6) How does the method compare with "Mind the GAP" and "StyleCLIP," which employ similar loss mechanisms and concepts in domain adaptation?

---

> ### Author Response · Authors · 2023-11-18
> **To Reviewer bxKn - Part 1**
>
> Thank you for your insightful comments about our work. We have added additional results with highlighting in the Appendix. Here we provide a point-by-point response to the issues raised by you.
>
> **The results of more domains**
>
> Consistent with prior work for few-shot generative domain adaptation (like CDC[1], RSSA[2], and DoRM[3]), our experimental data encompasses both global style ($\textit{sketch}$ and $\textit{baby}$) and local attributes ($\textit{smile}$ and $\textit{sunglasses}$). The combination of these domains demonstrates the effectiveness of our method since they encompass all types of generative domain adaptation.
>
> To provide more comprehensive evidence of validity, we conduct additional experiments on $\textit{Raphael}$ and $\textit{Caricature}$ for both single and hybrid domain adaptation. As shown in Fig. 15 of Appendix, the results integrate the characteristics from multiple target domains and maintain robust consistency with source domain, which further demonstrates the effectiveness of our method.
>
> **The generalizability to other domains**
>
> To verify the generalizability of our method to other domains, we conduct experiments on church domain following prior work. We adapt the pre-trained generator from $\textit{LSUN Church}$ to $\textit{Van Gogh's house paintings}$, $\textit{haunted houses}$, and the combination of them. As shown in Fig. 16 of Appendix, the results acquire the corresponding style and showcase the preservation of good cross-domain consistency. This aligns with the results observed in the face domain.
>
> **The advantages and disadvantages of few-shot domain adaptation techniques with discrimination**
>
> Few-shot domain adaptation techniques with discrimination (like DoRM, DualStyleGAN and 3DAvatarGAN) excel in generating images closely resembling the style of the target domain as its discriminator tends to memorize training images. However, they suffer from the notorious issue of model collapse. Especially in few-shot scenarios, they easily overfits to the target images, compromising the diversity of generated images. Additionally, they are challenging to extend into an end-to-end hybrid domain adaptation approach. Their approach to hybrid domain often involves separately training multiple models and interpolating style codes, necessitating multiple model size and training time.
>
> **Why is hybrid domain adaptation necessary in these cases**
>
> Image editing indeed enables local adjustments resembling the original domain, like $\textit{smile}$ and $\textit{baby}$. However, in this scenario, our method holds several advantages.
>
> Firstly, our objective is to generate images with attributes of target domain while maintaining considerable diversity, which can be applicable in scenarios like data collection. Image editing, on the other hand, requires an original image as input, rendering it impractical for such applications.
>
> Moreover, hybrid domain adaptation has the capability to generate images from hybrid domain containing multiple target attributes like $\textit{baby with the style of sketch}$ or $\textit{smiling person with the style of sketch}$. Image editing, however, lacks the ability to perform such global stylistic modifications, limiting its broader applications.

---

> ### Author Response · Authors · 2023-11-18
> **To Reviewer bxKn - Part 2**
>
> **The editability before and after the domain adaptation**
>
> We conduct the editing on the images generated by original and adapted model for both single and hybrid domain adaptation. As shown in Fig. 17 of Appendix, the results indicate the adapted generator maintains similar editability like pose to the original generator. This verifies that the our method effectively preserves original generator's attributes.
>
> **Results in 3D GAN setting**
>
> We conducted experiments using the popular 3D-aware image generation method, EG3D[4]. Specifically, we replace the discriminator as we did in Fig. 2 for both single and hybrid domain adaptation. As shown in Fig. 18 of Appendix, we adapt the pre-trained generator from FFHQ to $\textit{sunglasses}$ and the hybrid of $\textit{sunglasses}$ and $\textit{smile}$ with 10-shot training images per domain. We can observe that the results effectively integrate the attributes and preserve the characters and poses of source domain.
>
> **Comparisons with $\textit{Mind the GAP}$ and $\textit{StyleCLIP}$**
>
> While Mind the GAP has a similar loss with direction loss, one term of our proposed directional subspace loss, their motivations differ. Mind the GAP is proposed for one-shot domain adaptation, thus representing the entire target domain using the embedding of a single image. In contrast, for the few-shot setting, we propose representing the entire domain using the subspace formed by the embeddings of multiple images. Furthermore, we utilize pre-trained encoders to obtain separated subspaces corresponding to different domains, enabling us to accomplish hybrid domain adaptation.
>
> The direction loss in StyleCLIP is computed based on the cosine similarity between the generated images and textual prompts within the CLIP embedding space. It does not depend on the source image or domain. In practice, this loss leads to adversarial solutions and sees no benefit from maintaining diversity as depicted in Style-NADA[5]. A mode-collapsed generator producing only one image may be the best minimizer for the distance to a given textual prompt. Differently, our direction loss aims to preserve more characteristics from source domain and maintains its diversity.
>
> [1] Few-shot image generation via cross-domain correspondence. In Proceedings of the IEEE/CVF Conference on Computer Vision and Pattern Recognition, pp. 10743–10752, 2021.
>
> [2] Few shot generative model adaption via relaxed spatial structural alignment. In Proceedings of the IEEE/CVF Conference on Computer Vision and Pattern Recognition, pp. 11204–11213, 2022.
>
> [3] Domain re-modulation for few-shot generative domain adaptation. arXiv preprint arXiv:2302.02550, 2023.
>
> [4] Efficient geometry-aware 3d generative adversarial networks. In Proceedings of the IEEE/CVF Conference on Computer Vision and Pattern Recognition, pp. 16123–16133, 2022.
>
> [5] Stylegan-nada: Clip-guided domain adaptation of image generators. arXiv preprint arXiv:2108.00946, 2021.

---

> > ### Author Response · Authors · 2023-11-23
> > **Happy to provide additional clarifications**
> >
> > We hope these additional results and discussions can address your concerns. Please let us know if there are any further clarifications that we can offer. We would love to discuss more if any concern still remains.

---

> > > ### Comment · Reviewer_bxKn · 2023-12-03
> > >
> > > Thanks for the comments. I am still not convinced by the image editing argument the authors provided. Editing can be applied to a generated image. In the StyleGAN domain, the same edit can be applied to multiple generated images to render the same edit. The relevant papers I discussed can apply these edits after the domain adaptation, so I am not able to assess what advantage the method provides in this aspect. Hence, I would maintain my rating.

---

### Official Review · Reviewer_2zhA · 2023-11-01

**Soundness:** 3 good
**Presentation:** 4 excellent
**Contribution:** 3 good
**Rating:** 8
**Confidence:** 4

**Summary:**

Studies the problem of adapting an image generator (GAN) to additionally incorporate the attributes of multiple (hybrid) target domains with access only to individual target domain data. This is accomplished by optimizing a directional subspace loss that blends target attributes without mode collapse. Presents results on several generative DA benchmarks.

**Strengths:**

– The proposed problem – hybrid generative DA –  is interesting, and well-motivated. It is plausible that it might be easier to collect data for several independent target domains rather than for each possible combination thereof.

– The paper is well-written and very easy to follow. The presented qualitative results are compelling and experimental results are convincing, with strong+ and recent baselines such as DoRM

– The proposed method is simple, intuitive, and easy to follow

**Weaknesses:**

– I’m unsure about why this paper claims to introduce Hybrid generative DA, after acknowledging that prior work studying this problem exists (eg. DoRM). I understand that DoRM necessitates training target-specific models, but do they not also present some results on hybrid generative DA (eg. Figure 1 of their paper)?

– The approach requires manually tuning $\alpha_i$, and does not provide a principled method for hyperparameter selection. How sensitive is the approach to this parameter?

– The paper would be strengthened by a discussion and additionally, a comparison to recent methods for diffusion model personalization (eg. Dreambooth). While the methods would not necessarily be directly comparable, it would be a valuable point of reference

– The paper would benefit from a more detailed conceptual comparison to Nitzan et al., CVPR 2023: particularly, their approach of repurposing dormant directions in latent space for a new domain, seems quite related to the proposed approach

– The training time comparisons in Table 2 lack sufficient details: were the baseline numbers obtained by running publicly available code? Is it a completely apples-to-apples comparison, i.e. have all necessary parameters (eg. number and type of GPUs) been controlled for?

**Questions:**

Please address the weaknesses listed above.

---

> ### Author Response · Authors · 2023-11-18
> **To  Reviewer 2zhA**
>
> Thank you for your insightful comments about our work. We have added additional results with highlighting in the Appendix. Here we provide a point-by-point response to the issues raised by you.
>
> **Why we claim to introduce Hybrid generative DA**
>
> Prior techniques like style-mixing [1] can generate images from hybrid domain via interpolating latent codes. DoRM is just one of those methods. However, they lack a detailed investigation of HDA, which is merely mentioned as an additional feature. For example, they fail to provide systematic definition and proper evaluation metrics to distinguish good from bad.
>
> With regarding to the method in achieving HDA, they encounter two primary issues:
>
> (1) DoRM is not an end-to-end pipeline to address HDA. They primarily focus on single domain adaptation, necessitating the separate training and interpolating multiple models to accomplish HDA. This may not be the right way to approach this task.
>
> (2) DoRM necessitates intricate tuning of hyperparameters that interpolate between the source and target domains. As demonstrated in the DoRM paper, they assign a weight of 0.005 to the $\textit{baby}$ domain during single domain adaptation. Varying these hyperparameters significantly affects the outcomes. This complexity and sensitivity in parameter tuning make HDA's tuning more intricate and demanding.
>
> **How sensitive is our method to the domain coefficient $\alpha$**
>
> In our method for hybrid domain adaptation, this parameter controls the composition ratio of the attribute from each domain. As depicted in A.4 of Appendix, we use $\alpha_i = 0.5$ for most experiments without the need for complex and intricate adjustments. To further explore the sensitivity, we conduct the study for simple traversal of $\alpha_i$. As shown in the Fig. 14 of Appendix, the attributes of generated images transit smoothly between domains. Our method produces the similar attribute blending effect when $\alpha_i \in$ \{0.4, 0.5, 0.6\} .
>
> **Comparison to methods for diffusion model personalization**
>
> Current trend in customized text-to-image models like DreamBooth aim to mimic the appearance of subjects in a given reference set. Similar to ours, DreamBooth fine-tunes a pre-trained generator for the personalization. However, there are two key differences between personalization and HDA.
>
> (1) DreamBooth aims to retain the individuals from the training images, requiring similar individuals across the training set. Differently, our goal is to acquire the attributes of target domain like $\textit{baby}$ and preserve consistency with the source generator, ensuring that images generated from the same noise exhibit similar individuals.
>
> (2) Dreambooth utilizes text-to-image generator which necessitates intricate and laborious adjustments of prompts to synthesize images with specific attributes. Additionally, certain attributes are challenging to accurately describe using text, such as artistic paintings. Conversely, our adapted model adeptly preserves the domain-specific attributes of reference images without the need for intricate prompt engineering.
>
> **Conceptual comparison to Domain Expansion**
>
> Domain Expansion aims to expand the pre-trained generator to have the capacity to generate images from multiple domains. To the end, it proposes to repurpose dormant directions in latent space for the new domains. While our approach involves training images from multiple target domains, our objective is to create a unseen composition of given target domains that integrates attributes from them. For example, given training images from $\textit{baby}$ and $\textit{sketch}$, Domain Expansion aims to expand the generator to have the capacity to generate images from $\textit{baby}$ or $\textit{sketch}$. Differently, we aim to adapt the generator to unseen hybrid domain $\textit{baby with the style of sketch}$.
>
> **Details of the training time comparisons**
>
> In Table 2, our comparison about training time is fair. Specifically, we measure all the time on a single NVIDIA TITAN GPU. Due to the absence of open-source code for DoRM, we implement it following their description in the paper, based on the official implementation of StyleGAN2-ADA. The Seq method is also developed in a similar manner. As for StyleNADA, we utilize their open-source code for training. For an apples-to-apples comparison, we set the same batch size as 4 and the same resolution as $256 \times 256$.
>
> [1] Analyzing and improving the image quality of stylegan. In Proceedings of the IEEE/CVF conference on computer vision and pattern recognition, pp. 8110–8119, 2020.

---

> > ### Comment · Reviewer_2zhA · 2023-11-22
> > **Thank you for the detailed rebuttal**
> >
> > I am satisfied with the author response that addresses all of my concerns. I will raise my rating.

---

> > > ### Author Response · Authors · 2023-11-23
> > > **Thanks for your reply!**
> > >
> > > We are glad that our responses solve your concerns. Thank you again for your valuable feedback and suggestions!

---

### Official Review · Reviewer_fvXy · 2023-11-01

**Soundness:** 3 good
**Presentation:** 3 good
**Contribution:** 3 good
**Rating:** 8
**Confidence:** 3

**Summary:**

The paper introduces the few-shot generative Hybrid Domain Adaptation (HDA) task for image-based data. It is assumed that an image generator is accessed from a source domain and then from several target domains. Adaptation is performed on all target domains. To perform HDA, the paper presents a discriminator-free approach with a style GAN pre-trained on a source domain. It is adapted to individual target domains using a few shots from each domain and generates an image style showing composite attributes of each target domain separately.  The proposed approach relies on features extracted from standard transformer-based architectures. In addition, distance and direction losses are used to guide the generator to produce images with integrated features of the target domains while preserving the features of the source domain. Experiments are conducted using the FFHQ dataset and three evaluation metrics are used to measure performance and compare with different baselines.

**Strengths:**

- The paper is well written and easy to understand. The method is well described and the experiments are clearly presented.

- The discriminator-free approach presented is interesting as it reduces the computational cost of fitting compared to existing approaches.

- The approach shows promising results for all metrics compared to the baselines on the FFHQ dataset.

**Weaknesses:**

- (Major) The paper does not provide strong arguments as to why hybrid domain adaptation is a meaningful task. It is not clear why the current approaches and benchmarks, e.g. domain adaptation and domain generalisation, are not sufficient to address the problem presented. These points need further elaboration to motivate the paper.

- (Major) results are reported using 10 shots, while it would be interesting to see results using different numbers of shots, e.g. 1-shot and 5-shot.

- It is unclear whether the methodology is applicable to the generation of conditioned image styles. For example, generating an image that is conditioned on one or more styles of the target domain, rather than an image that spans all attributes of the target domains. It could discussed how the proposed methodology generalises to conditional generation.

- A study of the effect of different pre-trained image encoders would add value to the paper.

**Questions:**

- How would the proposed generator compare with well-known generator models such as DALLE, Imagen and etc.?

---

> ### Author Response · Authors · 2023-11-18
> **To Reviewer fvXy**
>
> Thank you for your insightful comments about our work. We have added additional results with highlighting in the Appendix. Here we provide a point-by-point response to the issues raised by you.
>
> **Why is hybrid domain adaptation a meaningful task**
>
> Current generative domain adaptation approaches typically employ the discriminator to discern whether generated images belong to the target domain. When we require to generate images with integrated attributes, they need to collect images from the hybrid target domain (e.g., a $\textit{smiling}$ $\textit{baby}$ with the style of $\textit{sketch}$ in Fig. 1). However, these images tend to be more difficult to collect compared with single domain in real-world scenarios. In contrast, HDA only requires collecting data for several independent target domains rather than for each possible combination as mentioned by **Reviewer 2zhA**. Under such circumstances, HDA offers greater flexibility and versatility to adapt the generator to more composite and expansive domain.
>
> **Results using different numbers of shots**
>
> We perform experiments on lower shots (5-shot and 1-shot) as suggested using various datasets. As shown in Fig. 11 of Appendix, the results of 5-shot are close to those of 10-shot, which integrates the attributes and maintains the consistency with source domain. Although multiple attributes have been learned, the results of 1-shot exhibit relatively lower cross-domain consistency compared with 10-shot and 5-shot.  This is because when there is only one reference image per domain, the subspace in our directional subspace loss degenerates into a single point (see Fig. 2). Then distinct generated images corresponding to different noise tend to converge towards the same image in CLIP's embedding space, which comprises cross-domain consistency as depicted in Section 3.3.
>
> **Results of conditional generation**
>
> As shown in Fig. 12 of Appendix, we conduct the experiments on conditional generation. Specifically, we collect 10-shot images with $\textit{red hair}$ and $\textit{sunglasses}$. Then we use masks to separate these attributes and adapt the generator with masked images for $\textit{red hair}$ and $\textit{sunglasses}$ respectively. We can observe that the generated images possess the corresponding attribute for both single DA and hybrid DA. Simultaneously, these images also maintain consistency with source domain.
>
> **Effect of different pre-trained image encoders**
>
> As shown in Fig. 13 of Appendix, we conduct experiments on pre-trained Swin and Dinov2 to explore the impact of different image encoders on generated images. Their results all achieve HDA, indicating that our method is agnostic to different pre-trained image encoders. Although they exhibit slight stylistic differences, these are due to their different approaches to extract features into separated subspaces, as depicted in Fig. 8 of Appendix. To converge to the exact realization of the target domain, our method employs the ensemble technique that exploits both Swin and Dinov2. As shown in the figure, the results closely resembles the attributes of the target domain while maintaining the best consistency with source domain.
>
> **Compare with DALL-E and Imagen**
>
> DALL-E and Imagen are both text-to-image models known for their ability to generate images based on natural language descriptions. They exhibit the advantage of autonomously producing realistic images with specific attributes while maintaining considerable diversity. However, in comparison to our generator, they exhibit two primary drawbacks.
>
> (1) They necessitate intricate and laborious adjustments of prompts to generate images with specific attributes. Additionally, certain attributes are challenging to accurately describe using text, such as artistic paintings. Conversely, our generator can preserve the domain-specific attributes of reference images without the need for intricate prompt engineering.
>
> (2) It is hard to control the characters within images via prompts for them. In contrast, our adapted model maintains strong consistency with the source generator, preserving the same characters from the same noise.
>
> Besides, ours differs from the current trend in customized text-to-image models like DreamBooth. DreamBooth aims to retain the individuals from the training images, requiring similar individuals across the training set. Differently, our goal is to acquire the attributes of target domain like $\textit{baby}$ and preserve consistency with the source generator, ensuring that images generated from the same noise exhibit similar individuals.

---

> > ### Author Response · Authors · 2023-11-23
> > **Happy to provide additional clarifications**
> >
> > We hope these additional results and discussions can address your concerns. Please let us know if there are any further clarifications that we can offer. We would love to discuss more if any concern still remains.

---

### Author Response · Authors · 2023-11-22
**Welcome Further Response and Discussion**

Dear Reviewers:

We sincerely thank you for your great efforts in reviewing this paper. We are pleased to read that our proposed problem is **interesting** and **well-motivated** [2zhA], our method is **intuitive** and **effective** [fvXy, 2zhA], and our results are **compelling** [fvXy, 2zhA, bxKn].

We have tried our best to address all the mentioned concerns and problems. **Our manuscript has been revised** to include the changes according to all the reviewers’ insightful comments, making our research more robust and accessible.

**As the deadline for Author-Reviewer discussion is approaching**, we are eagerly looking forward to your responses. Please let us know if there are any additional clarifications or experiments that we could offer. We would love to discuss more if any concern still remains. Thanks again for your time!

Best,

Authors

---

### Meta-Review · Area_Chair_KG59 · 2023-12-05

**Metareview:**

This paper received contrasting scores, namely, 8, 8, and 5.
Two reviewers acknowledged the goodness of the work and convincingly voted for acceptance. The remaining reviewer (rev. bxKn) was instead not fully convinced about the work, and the rebuttal has not contributed to improving the situation.

A number of concerns was raised by the most positive reviewers, to which authors satisfactorily replied in their rebuttal.
The remarks raised by the most critical reviewer regarded the limited diversity of the addressed domains, and hence of the related validation, missing discussion and validation related to the few-shot nature of the considered problem, not clear justification of the approach when confronted with GAN-based algorithms (StyleGAN-like).
The authors wrote an extensive rebuttal, but the reviewer remained of the same opinion, especially in regard of the comparison with former StyleGAN-like methods that do the same job.

The AC acknowledges the effort of the authors in replying to all raised remarks and assesses favorably this work, also in light of the convincing evaluations of the majority of the reviewers. In the end, this paper can be accepted for publication to ICLR 2024.
In the final version, it is recommended to explain better the limitations of the approach in relation to the reviewers' comments, especially concerning the limited diversity of the addressed domains, which are addressing only faces and just one example of Church data: clearly identifying pros and limits of the proposed approach will surely be beneficial to a larger spread of the method.

**Justification For Why Not Higher Score:**

Good work but with no unanimous positive evaluation.

**Justification For Why Not Lower Score:**

Nevertheless, authors replied extensively to reviewers' remarks and the remaining issue from rev. bxKn does not seem corroborated by substantial ground. Authors tried to explain that their work acts differently, but the reviewer is sticking in his assessment.
I am more in agreement with the authors' justifications.
Moreover, rev. bxKn is declaring to be little knowledgeable about this work (confidence = 2).

---

### Decision · Program_Chairs · 2024-01-16

Accept (poster)